# Vena Cava Thrombus in Patients with Wilms Tumor

**DOI:** 10.3390/cancers14163924

**Published:** 2022-08-14

**Authors:** Clemens-Magnus Meier, Rhoikos Furtwängler, Dietrich von Schweinitz, Raimund Stein, Nils Welter, Stefan Wagenpfeil, Leo Kager, Jens-Peter Schenk, Christian Vokuhl, Patrick Melchior, Jörg Fuchs, Norbert Graf

**Affiliations:** 1Department of General Surgery, Visceral, Vascular and Pediatric Surgery, Saarland University Medical Center, 66421 Homburg, Germany; 2Department of Pediatric Oncology and Hematology, Saarland University Medical Center, 66421 Homburg, Germany; 3Department of Pediatric Surgery, Dr. von Hauner Children’s Hospital of the University of Munich, 80337 Munich, Germany; 4Center for Pediatric, Adolescent and Reconstructive Urology, University Medical Center Mannheim, Medical Faculty Mannheim, Heidelberg University, 68167 Mannheim, Germany; 5Institute for Medical Biometry, Epidemiology and Medical Informatics, Saarland University, Campus Homburg, 66421 Homburg, Germany; 6St. Anna Children’s Hospital, Department of Pediatrics, Medical University Vienna, 1090 Vienna, Austria; 7St. Anna Children’s Cancer Research Institute, 1090 Vienna, Austria; 8Pediatric Radiology Section, Department for Diagnostic and Interventional Radiology, University Hospital Heidelberg, 69120 Heidelberg, Germany; 9Institute of Pathology, University Hospital Bonn, 53127 Bonn, Germany; 10Department of Radiation Oncology, Saarland University Medical Center, 66421 Homburg, Germany; 11Department of Pediatric Surgery and Urology, University Hospital Tübingen, 72076 Tuebingen, Germany

**Keywords:** Wilms tumor, treatment, surgery, preoperative chemotherapy, vena cava thrombus

## Abstract

**Simple Summary:**

Wilms tumor (WT) is the most common solid renal tumor in childhood. Today, more than 90% are alive 5 years after treatment. If the tumor extends into the major abdominal vessels, treatment still faces significant challenges. In these cases, tumor removal from the vessel is also required, sometimes requiring complex management. Knowledge of factors that positively or negatively affect treatment and survival is therefore of particular interest. We, therefore, compiled data on this small group of WT patients and evaluated them in terms of survival. This revealed that certain tumor-related features, as well as the presence or absence of metastases, significantly affect survival. This information helps us to further improve the treatment for this group of patients.

**Abstract:**

(1) Background: Vena cava thrombus (VCT) is rare in Wilms tumor (WT) (4–10%). The aim of this study is to identify factors for an outcome to improve treatment for better survival. (2) Methods: 148/3015 patients with WT (aged < 18 years) and VCT, prospectively enrolled over a period of 32 years (1989–2020) by the German Society for Pediatric Oncology and Hematology (SIOP-9/GPOH, SIOP-93-01/GPOH and SIOP-2001/GPOH), are retrospectively analyzed to describe clinical features, response to preoperative chemotherapy (PC) (142 patients) and surgical interventions and to evaluate risk factors for overall survival (OS). (3) Results: 14 VCT regressed completely with PC and another 12 in parts. The thrombus was completely removed in 111 (85.4%), incompletely in 16 (12.3%), and not removed in 3 (2.3%). The type of removal is unknown in four patients. Patients without VCT have a significantly (*p* < 0.001) better OS (97.8%) than those with VCT (90.1%). OS after complete resection is (89.9%), after incomplete (93.8%) and with no resection (100%). Patients with anaplasia or stage IV without complete remission (CR) after PC had a significantly worse OS compared to the remaining patients with VCT (77.1% vs. 94.4%; *p* = 0.002). (4) Conclusions: As a result of our study, two risk factors for poor outcomes in WT patients with VCT emerge: diffuse anaplasia and metastatic disease, especially those with non-CR after PC.

## 1. Introduction

Wilms tumor (WT) is the most common solid renal tumor in childhood. Treatment of nephroblastoma has made great improvements since the 1970s [1,2,3,4]. Today, the 5-year survival rate is greater than 90% [5,6,7,8]. This success has been achieved through prospective, randomized, multicenter trials conducted by the International Society of Pediatric Oncology (SIOP), the Children’s Oncology Group in North America (COG), and others, which allow better treatment stratifications based on individual patient risk factors. However, there are subgroups for whom treatment still faces significant challenges. This includes the extension of the tumor into the vena cava (VC). Intracaval tumor thrombus is found in 4–10% of cases, ranging in extent from infrahepatic to intracardiac [9,10,11,12]. In these cases, tumor removal from the vessel is also required, sometimes requiring complex management.

Successful treatment depends mainly on histological type and stage, with upstaging in case of incomplete removal of the tumor. Other factors for the outcome are age, response to chemotherapy as well as molecular findings. In addition to surgery and chemotherapy, radiation therapy is needed in only about 20% of all cases today [13,14]. These principles apply to all patients, irrespective of whether they have undergone primary surgery (PS) or preoperative chemotherapy (PC) [15]. 

Based on data from the SIOP/GPOH trials and studies between 1989 and 2020, the aim of this study is to illustrate the procedure when the tumor extends to the VC and to discuss the implications for long-term outcomes.

## 2. Materials and Methods

This retrospective analysis is based on data from three successive trials of the German Society of Pediatric Hematology and Oncology (GPOH) (SIOP 9/GPO, SIOP 93-01/GPOH, and SIOP 2001/GPOH) for the treatment of children and adolescents with kidney tumors. These studies include prospectively enrolled patients from Germany, Austria, and Switzerland between 1989 and 2020 (up to 1 August 2020). All studies were reviewed and approved by the Ethics Committee of the Saarland Medical Association (/LS from 23.04.1993, no. 136/01 from 20 September 2002, and 248/13 from 13 January 2014). 

Using a standardized query, the following data were obtained from the studies’ database: demographic data, radiological findings, kind of treatment start (PS or PC), surgical procedure, postoperative radiation, tumor histology, completeness of tumor, and thrombus removal, follow-up and survival data. Data on surgical procedures, postoperative complications as well as other clinical, pathological, and outcome data were anonymized before data analysis. For certain questions, patients without cava thrombus served as a control group.

Diagnosis of vena cava thrombus (VCT) was either based on imaging studies at diagnosis and/or after preoperative chemotherapy or on surgical findings of a VCT when imaging studies were not available. Classification of tumor and thrombus extension into the VC was performed by computed tomography (CT) or magnetic resonance imaging (MRI). Extension into the vena cava was classified as follows: none, infrahepatic, retrohepatic, suprahepatic, and intracardiac. Pairs of radiological imaging (at diagnosis, after PC) were used to evaluate the influence of chemotherapy on the development of the thrombus for those who received PC.

Statistical analysis was performed using SPSS 27 for Mac (IBM SPSS Statistics 27, 1 New Orchard Road, Armonk, NY 10504-1722, USA). χ^2^-test and Fischer exact test were used to compare relative frequencies between independent groups. Values not showing normal distribution were compared using Mann–Whitney-U tests. To compare paired variables, the Wilcoxon test was used. Kaplan–Meier analyses were used to assess survival for different questions. The log-rank test was used to test for significance. Two-sided significance was defined as *p* < 0.05 for all tests.

## 3. Results

From 1989 to 2020, 3710 patients with renal tumors were registered in consecutive SIOP/GPOH studies. Resulting in a total of 3015 patients with WT aged up to 18 years. In 148 patients, inferior vena cava thrombus was diagnosed in unilateral 140 (94.6%) and in bilateral WT 8 (5.4%). A total of 142 of them received primary chemotherapy, while 5 underwent initial surgery. In one patient, the primary treatment is unknown.

Radiological data are available for 118/148 patients with VCT at the time of diagnosis. A tumor thrombus extending into the VC was found in 115 cases (97.5%). In two-thirds of cases (66.1%, *n* = 78), the tumor thrombus was localized infrahepatic. Retrohepatic extension was seen in only 3.4% of patients (*n* = 5), suprahepatic and intracardiac extension in 8.1% (*n* = 12) and 13.1% (*n* = 20). VCT was not seen in 3 patients (2%) at diagnosis but later at preoperative imaging. In 30 patients, VCT was detected at the time of surgery.

Basic data of the cohorts are given in Appendix A. Compared to patients without VCT, patients with thrombus are significantly older (no VCT median 39 (21/62) months: VCT median 56 (35/74) months, *p* < 0.001), have larger tumor volumes (no VCT ≥ 500 mL 1031/2664, 38.7%: VCT ≥ 500 mL 72/141, 51.1%, *p* = 0.004), and more often have a low-risk (LR) and less often a high-risk (HR) tumor (no VCT: LR 92/2867, 3.2%, intermediate-risk (IR) 2333/2867, 81.4%, HR 442/2867, 15.4%: VCT: LR 17/148, 11.5%, IR 119/148, 80.4%, HR 12/148, 8.1%, *p* < 0.001). In addition, histologic types of IR are different between no VCT and VCT (*p* < 0.001); for example, more patients with VCT are affected with completely necrotic and regressive types and fewer with mixed type. (Table 1).

Patients with VCT are significantly more likely to have metastatic disease at diagnosis (no VCT 436/2823, 15.4%: VCT 81/148, 57.7%, *p* < 0.001), within the lung (no VCT 405/2867, 14.1%: VCT 78/148, 52.7%, *p* < 0.001) and liver (no VCT 54/2867, 1.9%: VCT 22/148, 14.9%, *p* < 0.001) being most affected. In addition, local stages are higher in patients with VCT (stage I 12,2% vs. 60,7%; stage II 36,7% vs. 21.7%; stage III 51.0% vs. 17.6%; *p* < 0.001). Despite these differences, the outcome of patients with and without VCT is statistically not significant (Figure 1).

Lymph nodes are more likely to be pathological enlarged or even infiltrated at surgery (no VCT: normal 1606/2619, 61.3%, pathological enlarged 947/2619, 36.2%, infiltrated 66/2619, 2.5%: VCT normal 69/138, 50%, pathological enlarged 62/138, 44.9%, infiltrated 7/138, 5.1%, *p* < 0.001). 

According to the study protocols, patients received actinomycin D (45 µg/kg) and vincristine (1.5 mg/m^2^). In the case of metastatic disease, Doxorubicin (50 mg/m^2^) was added. To assess the effect of chemotherapy on tumor thrombus, 113 pairs of data with radiologic imaging could be used. PC shortened the thrombus in 26 cases (*n* = 26/113, 23%) according to the classification levels used, but it did grow in one patient (*n* = 1/113, 0.9%). In the remaining 86 cases (*n* = 86/113, 76.1%), there was no change in VCT classification. A detailed analysis of changes in thrombus level is given in Table 2. The one patient with progressive thrombus growth had a suprahepatic level at diagnosis, which grew further into the right atrium. Tumor histology in this patient was diffuse anaplasia. In three other patients who showed no thrombus at diagnosis, a thrombus was found later in preoperative imaging or during surgery. The histology of these was two times stromal type and once complete necrotic. Appendix A shows the distribution of changes with respect to tumor histology. A reduction in thrombus size is found in all histologic types except anaplasia. None of the patients with focal anaplasia showed any improvement. Most changes are found in the regressive type. The changes found between histologic types or risk groups are significantly different. There was no statistically significant difference in patients with (18/64, 28.1%) and without metastatic disease (11/49, 22.4%). 

Surgical reports about intraoperative findings were available in 2835 cases (*n* = 2835/3015, 94%). VCT was found in 130/148 cases during tumor resection. In 14/148 patients, the VCT did disappear completely, as shown by imaging studies. In 4/148, VCT was diagnosed by imaging, but surgical data are missing. Thrombus was completely removed in 111 cases (*n* = 111/130, 85.4%), incompletely in 16 (*n* = 16/130, 12.3%), and not removed in another 3 patients (*n* = 3/130, 2.3%). Cardiopulmonary bypass was required in 10 patients (*n* = 10/130, 7.7%), and intracardiac thrombus was present in 8 of these cases. In 13 cases (*n* = 13/130, 10%), the VC was replaced by a prosthesis. In one of the cases, the thrombus was previously unknown. In 40 cases (*n* = 40/130, 30.8%), surgery was performed in collaboration with a vascular surgeon. Detailed analysis of 123 patients with a preoperatively known level of thrombus showed that the majority of infrahepatic thrombi (*n* = 76/123, 82.6%) could be completely removed. A total of 12 of 15 intracardiac thrombi were also completely removed, only 3 incompletely. (Table 3) The three thrombi that were based on the surgical decision not removed were all located infrahepatically. 

Data on status after preoperative chemotherapy were available from 58 of the 81 patients (71.6%) with metastases at diagnosis. A total of 27 (46.6%; 19 after PC and 8 after PC and additional surgery) achieved complete remission. A total of 31 (53.4%) did not achieve remission because of incomplete metastasectomy or inoperability. According to the postoperative pathological findings, local stage I was found in 18 (*n* = 18/147, 12.2%), stage II in 54 (*n* = 54/147, 36.8%), and stage III in 75 (*n* = 75/147, 51%) patients. In one patient local stage is unknown. This resulted in 67 patients (*n* = 68/147, 46.3%) receiving local irradiation (local stage III without complete necrosis and, in addition, diffuse anaplasia in local stage II). This resulted in a general staging of stage I 8 (5.4%), II 29 (19.6%), III 30 (20.3%), and IV 81 (54.7%). 

Survival data are available from 2830 patients (*n* = 2830/3015, 93.9%). A total of 6/148 patients with VCT were excluded from analysis as survival is unknown. During the observation period of more than 30 years (mean 11.6 ± 8.5 years, median 10.8 (3.9/17.7) years), 58 patients without VCT (*n* = 58/2688, 2.2%) and 14 with VCT (*n* = 14/142, 9.9%) died (*p* < 0.001). No patient in the VCT group died during or as a complication of surgery. The causes were tumor progression in 12 cases (2 progression of pulmonary metastases, 1 respiratory failure,1 multiorgan failure, 8 progression of tumor disease with development of further metastases), non-tumor or treatment-related in one patient, and the cause is unknown in another patient.

Overall survival (OS) of patients with thrombus in the inferior vena cava after complete resection is 89.9%, and after incomplete resection (93.8%). None of the patients died in whom the thrombus was not removed. The differences between the procedures are not significant (*p* = 0.794). (Figure 2A) Kaplan–Meier analysis further shows that a reduction in thrombus size by preoperative chemotherapy has no effect on survival (*p* = 0.749). (Figure 2B) In contrast, the histologic risk group significantly affected survival (*p* < 0.001). While no patient died in the low-risk group, OS was 93% in the intermediate-risk group and was significantly worse in the high-risk group. Half of the patients died (50%). (Appendix A) If the low and intermediate-risk groups are taken together, significantly more patients died (*p* < 0.001) if they were classified as high risk, regardless of a VCT (Appendix A).

The group of patients without a VCT had a significant better (*p* < 0.001) overall survival (*n* = 2630/2688, 97.8%) than those with a VCT (*n* = 128/142, 90.1%). (Figure 2C) However, the poorer outcome of patients with VCT is solely based on metastasis. In patients with localized disease, no patient with a VCT has died so far. Therefore, the outcome of patients with VCT is correlated with metastatic disease (Appendix A) and anaplasia (Appendix A). We also compared all patients with VCT and anaplasia together with the stage IV patients who did not achieve remission by preoperative chemotherapy with the remaining patients. It shows a significantly worse outcome (*p* = 0.002) for the first group (Figure 2D). In addition, the overall survival of patients with diffuse anaplasia and no remission of metastasis after preoperative chemotherapy is different (*p* = 0.078) between those with VCT (35 patients, OS = 77.1% after 5 years) compared to those without VCT (196 patients, OS = 87.8% after 5 years) (Figure 3).

## 4. Discussion

Continuous extension into the vena cava with the formation of a thrombus is a common finding in Wilms tumor. In our large cohort of 3015 Wilms tumor patients, we found 148 cases (*n* = 148/3015, 4.9%) with tumor thrombus in the inferior vena cava. This makes this study one of the largest published evaluations of patients with VCT in the last 20 years. The frequency itself is in line with the data of other studies [9,10,11,12] but lower compared to the 6% published by Shamberger [16] or 8.1% by Lall [17].

A combination of MRI / CT and ultrasound has been used to assess tumor extension at diagnosis as well as during follow-up. Because of its reproducibility, CT and today, MRI is chosen as the standard procedure as it allows tumor staging, monitoring chemotherapy, and preoperative planning by volume rendering and three-dimensional postprocessing. MRI has special significance in assessing the extent of the tumor, especially in the inferior vena cava, because of its excellent visualization of the soft tissue [18,19]. MRI shows additional benefits compared to the ultrasound, as McDonald reports. In almost every second patient (*n* = 19/40) of their study, additional findings led to a change of local stage and treatment [20]. In this respect, tomographic imaging allows correct staging, and undertreatment is avoided [21]. Doppler ultrasonography is an easy-to-use technique that additionally provides great benefit in the detection of intravascular thrombi [22].

Five different levels are defined in our study protocol to classify the extent of the thrombus: none, infrahepatic, retrohepatic, suprahepatic, and intracardiac. These are radiologically determined levels according to the upper extent of the tumor thrombus. Alternative classifications have been proposed by Staehler [23], Pritchett [24], and Hinman [25]. In our cohort, two-thirds of patients (67.8%) showed infrahepatic VCT comparable to other studies. For example, Lall et al. reported VCT at infrahepatic level in half of their patients (*n* = 26/59, 44.1%). Our frequency of intracardiac VCT of 17.4%, is also comparable to others such as Schamberger (*n* = 31/164, 18.9%) [16], Lall 10 (*n* = 10/59, 16.9%) [17] and other reports [16,17,22].

Patients with VCT are significantly older and have larger tumors than those without VCT, which is also shown in other reports for age [17,26,27] and size [26]. VCT was found more frequently in right-sided tumors (56%), which has also been reported previously. Here, the shorter renal vein on the right side compared to the left side is seen as the cause [17,22,27,28,29]. Furthermore, metastatic disease is already present in more than half of the cases. Lung and liver metastases are particularly noteworthy here. Metastatic disease in VCT is also seen by others [12,27], whereas the commonest site for metastasis was the lungs [12,27] and liver [12]. This general baseline data indicate that the disease is already advanced in patients with VCT.

According to the protocol, most patients (*n* = 142/148, 95.9%) received preoperative chemotherapy. The reasons for primary surgery were emergency surgery in one case, uncertain diagnosis in two cases, and unknown reasons in two other cases. Analysis of imaging data allowed us to evaluate the change in classification level of the thrombus due to chemotherapy. PC shortened the thrombus by 23%. Similar results were found in other studies, where even higher rates of thrombus shrinkage of 45–80% [16,17,26,30,31] and in all patients [32] were reported. A total of 14 VCT (11.9%) regressed completely. Elayadi also reported complete regression of 18 thromboses (*n* = 18/48, 37.5%). Most (*n* = 16/48, 33%) were infrahepatic, as in our data [27]. Reported rates of complete regression are ranging from 11% to 47% [26,27,30]. In addition, 11 patients with higher levels showed shrinkage of the thrombus. The number of intracardiac thrombi decreased by 30%, consistent with Hadley’s or Elayadi’s reports [27,30]. Further growth was seen in only one patient (0.9%) with anaplasia. In 86 cases (76.1%), there was no change in classification. Others also report the absence of shrinkage, for example, Elayadi in 50% of patients [27].

Compared to the group of non-VCT patients, more low and less high-risk types are found in our cohort. The completely necrotic type is three times more frequent, and the regressive type accounts for almost 42.6% (non-VCT: 27.7%). These results are consistent with those of other studies reporting a low number of anaplasia compared with the favorable pathologic subtype [16,17,22,27]. Considering the histological type of tumor, thrombus shrinkage is greatest in the regressive type (64%), but it also occurs in all other types except anaplasia.

VCT was removed in almost all cases (*n* = 127/130, 97.7%) where a VCT was found during surgery. The thrombus was completely removed in 111 cases (85.4%), incompletely in 16 (12.3%), and not removed in another 3 patients (2.3%). The reasons why VCTs were not removed in the three cases were due to the surgeon’s opinion that the VCT was not removable. These results are comparable to the data of Schamberger, who reported a rate of 73% (*n* = 120/164) of completely resected VCT. The number of VCTs that were not resected was even higher in his study with 18 (*n* = 18/164, 11%) [16]. All but one VCT was removed in Elayahi’s study. This one was intracardiac. Since no cardiopulmonary bypass was possible, the VCT was left in place and then irradiated [27].

In all our 15 patients still having intracardiac thrombus after preoperative chemotherapy, the thrombus was removed. Completely in 12 of 15 and incompletely in three. Cardiopulmonary bypass was used in eight procedures (*n* = 8/15 53.3%). This is less than in Schamberger’s data with 82.4% (*n* = 14/17) [16].

Overall survival (OS) of patients with thrombus in the vena cava inferior after complete resection is 89.9% compared to 93.3% after incomplete resection of VCT. Upstaging in case of incomplete resection with additional radiotherapy after surgery may overcome a negative impact on survival after incomplete resection of VCT. This assumption is also supported by Boam [33]. Histologic risk group significantly affected survival. While no patient died in the low-risk group, OS was 93% in the intermediate-risk group, and OS was significantly worse in the high-risk group, half of the patients died. Similar results are found for the patients without VCT. Thus, patients with VCT do not perform worse here. However, when comparing the OS of a combined group of low and intermediate-risk patients with (93.8%) or without VCT (98.6%), the OS of patients with VCT and high risk is significantly worse (50%) than high risk and no VCT (92.3%). In Schamberger’s report, the 3-year survival rate for children with favorable histology was 90%, and the rate for those with anaplasia was 41.7% [16]. In Lall’s study cohort, all three patients with VCT and unfavorable histology died (*n* = 3/3, 100%), whereas the survival rate of the 56 patients with favorable histology was 76.8% (*n* = 43/56) [17]. Similarly, Ritchey reports 3-year survival of 86% with favorable and 35% with unfavorable histology [29].

Based on the study data, the whole group of patients without a VCT had a significantly better overall survival (97.8%) than those with a VCT (90.1%), which is also reported by Lall [17]. Shamberger’s data also show that the survival rate of patients with IVC is also lower (76.9%:80.3%). However, this does not reach significance in his analysis. Better survival is especially true for the group of patients who have only localized disease. If metastatic disease is present, there are no differences (*p* = 0.341) in survival between patients with and without VCT, but their survival is lower (81.8%) compared to patients with localized disease (85.7%). In addition, we could demonstrate that survival was significantly worse in patients with VCT and metastatic disease who did not achieve complete remission of metastasis after PC. This means that the difference in survival in patients with or without VCT is triggered by metastasis. This is underlined by the fact that no patient without metastases but with VCT died.

Taking histology and metastatic disease into consideration, especially those patients with diffuse anaplasia and without remission of metastasis after PC are a high-risk group for treatment failure. This is a new finding. In larger studies, for example those of Schamberger (*n* = 165) [16], Lall (*n* = 59) [17], or Elayadi (*n* = 51) [27], such a risk group is not defined.

## 5. Conclusions

Tumor thrombus in the vena cava occurs in only a small proportion of Wilms tumor patients. In most cases, they are asymptomatic. Preoperative chemotherapy can induce shrinkage of the thrombus, which facilitates resection. Nevertheless, removal of the thrombus is a complex and high-risk procedure, sometimes involving cardio-pulmonaly bypass or vascular replacement. Although today surgery-related mortality is low, OS is worse compared to patients without VCT. As a result of our study, two risk factors for poor outcomes in WT patients with VCT emerge: diffuse anaplasia and metastatic disease, especially those with no remission after PC.

## Figures and Tables

**Figure 1 cancers-14-03924-f001:**
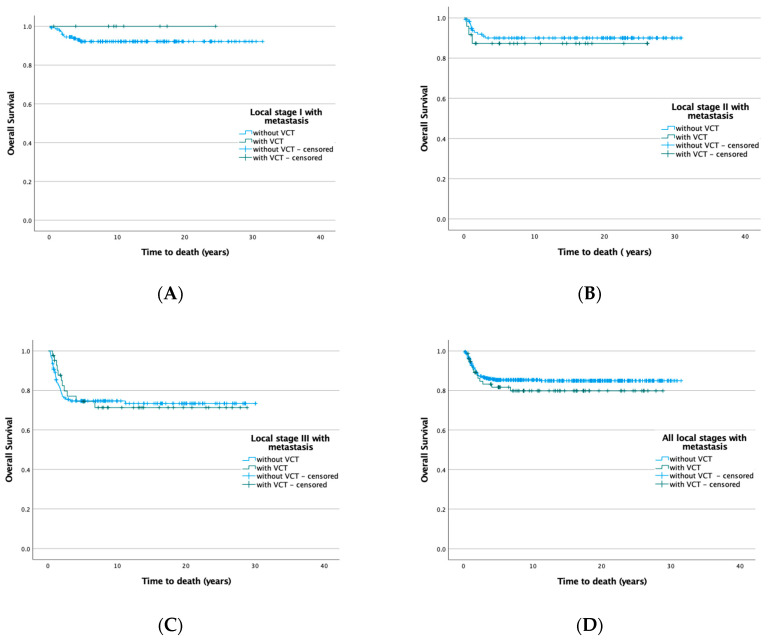
Overall survival with and without VCT depending on local stages in patients with metastatic disease. (**A**) Local stage I (no VCT 92.6%, VCT 100, *p* = 0.424); (**B**) local stage II (no VCT 90.5%, VCT 87.5, *p* = 0.631); (**C**) local stage III (no VCT 74.8%, VCT 74.4, *p* = 0.961); (**D**) all stages (no VCT 85.7%, VCT 81.8, *p* = 0.341).

**Figure 2 cancers-14-03924-f002:**
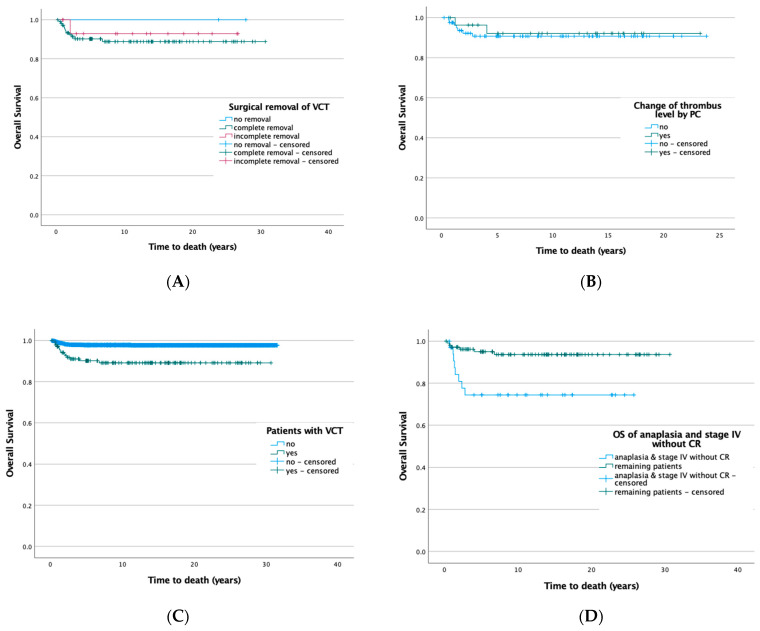
(**A**) OS by surgical approach (complete 89.8%, incomplete 93.8%, or no removal 100%, log rank: *p* = 0.794). (**B**) Influence of change in thrombus size on OS (yes 92.9%, no 91.6%, log rank: *p* = 0.749). (**C**) OS of patients with and without VCT (no 97.8%, yes 90.1%, log rank: *p* < 0.001). (**D**) Overall survival of patients with VCT and anaplasia or no CR after PC and surgery in stage IV (1: *n* = 35/142) versus the remaining patients with VCT (2: *n* = 107/142). (OS 1: 77.1%, OS 2: 94.4%, log rank: *p* = 0.002).

**Figure 3 cancers-14-03924-f003:**
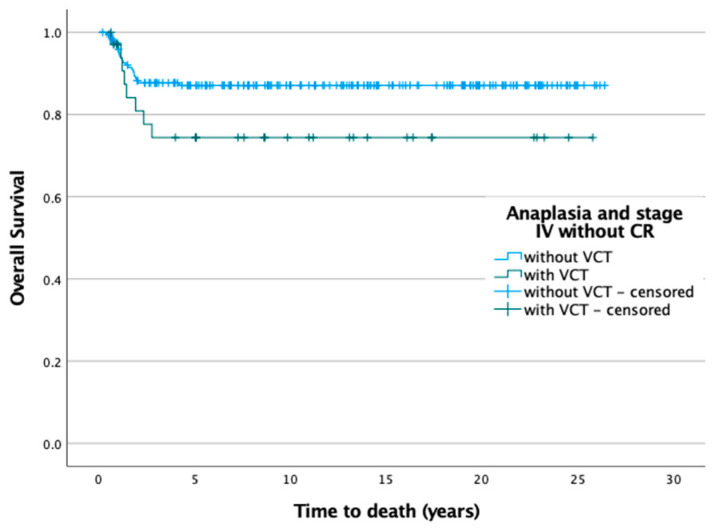
Overall survival of patients with diffuse anaplasia and no remission of metastasis after preoperative chemotherapy depending on VCT.

**Table 1 cancers-14-03924-t001:** Distribution of vena cava thromboses according to histology (*p* < 0.001) at time of diagnosis.

Histological Type	No Vena Cava Thrombus		Vena Cava Thrombus		All	
	*n*	%	*n*	%	*n*	%
Completely necrotic type	92	3.2	17	11.5	109	3.6
Epithelial type	196	6.8	6	4.1	202	6.7
Stromal type	279	9.7	16	10.8	295	9.8
Mixed type	1010	35.2	32	21.6	1042	34.6
Regressive type	795	27.7	63	42.6	858	28.5
Focal anaplasia	53	1.8	2	1.4	55	1.8
Blastemal type after PS	92	3.2	0	0.0	92	3.1
Blastemal type after PC	224	7.8	7	4.7	231	7.7
Diffuse anaplasia	126	4.4	5	3.4	131	4.3
All	2867	100	148	100	3015	100

**Table 2 cancers-14-03924-t002:** Alteration of the thrombus level by chemotherapy (*n* = 118). The first row shows the level of thrombus of each group at the time of diagnosis. The second block shows the levels after PC for the individual groups. The third block summarizes the changes.

At Diagnosis	Level of Thrombus	None		Infrahepatic		Retrohepatic		Suprahepatic		Intracardiac		∑	
		3		78		5		12		20		118	
**After Chemotherapy**	No thrombus	-	-	13	16.7%	1	20%	-	-	-	-	14	11.9%
	Infrahepatic	2	66%	63	80.7%	2	40%	2	16.7%	2	10%	71	60.2%
	Retrohepatic	-	-	-	-	2	40%	2	16.7%	1	5%	5	4.2%
	Suprahepatic	-	-	-	-	-	-	7	58.3%	3	15%	10	8.5%
	Intracardial	-	-	-	-	-	-	1	8.3%	14	70%	15	12.7%
	Unknown	1	33%	2	2.6%	-	-	-	-	-	-	3	2.5%
	No thrombus	-	-	13	16.7%	1	20%	-	-	-	-	14	11.9%
	Thrombus	3	100%	65	83.3%	4	80%	12	100%	20	100%	104	88.1%

**Table 3 cancers-14-03924-t003:** Results of surgical resection of the thrombus out of the inferior vena cava (*n* = 123, *p* = 0.832). CPBDS cardiopulmonary bypass during surgery, VCP vena cava prosthesis. Included in this analysis are only those patients with information about the level of the thrombus after preoperative chemotherapy and the kind of removal of the thrombus at surgery. Not included in the analysis are two patients with no information about the level of thrombus after preoperative chemotherapy.

Removal of Thrombus		None		Complete		Incomplete		All		CPBDS		VCP	
		*n*	%	*n*	%	*n*	%	*n*	%	*n*	%	*n*	%
**Level at surgery**	None	-	-	1	100	-	-	1	1.6	-	-	-	-
	Infrahepatic	3	3.3	76	82.6	13	14.1	92	74.2	1	10	6	50
	Retrohepatic	-	-	5	100	-		5	4	-	-	1	8.3
	Suprahepatic	-	-	10	100	-		10	8.1	1	10	2	16.7
	Intracardiac	-	-	12	80	3	20	15	12.1	8	80	3	25
**All**		3	2.4	104	84.6	16	13	123	100	10	100	12	100

## Data Availability

The data presented in this study are available on request from the corresponding author. The data are not publicly available due to ongoing analysis.

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
