# Peer review of "Vena Cava Thrombus in Patients with Wilms Tumor"

_cancers, 2022, doi:10.3390/cancers14163924_

Round 1

Reviewer 1 Report

This study is drawn from a large cohort of patients treated on the SIOP/GPOH studies from 1989 to 2020 and hence they received a sequence of three different treatment protocols during that time.  From 3,015 patients less than or equal to 18 years of age 148 were identified as having vena caval tumor thrombus, the majority were identified during imaging at diagnosis and a small number at subsequent tumor resection.   They demonstrated that some of these tumors regressed during neoadjuvant therapy.  I think the authors could simplify the results of tumor regression both in their text and in the accompanying tables.  The vast majority of children received neoadjuvant chemotherapy so the authors can not address the question as to whether there were fewer surgical complications with primary vs secondary surgery.  The "bottom line" conclusions of this review are that patients with vena caval tumor thrombus had a higher rate of survival if they did not have diffuse anaplasia histology or metastatic disease.   These findings are certainly not at all surprising.  This study supports the finding in prior studies, but does not really provide new information regarding this clinical situation.  

Author Response

We want to thank the reviewer for a detailed review of our paper. Here are reply to the different comments.

  1. "I think the authors could simplify the results of tumor regression both in their text and in the accompanying tables." We shortened the text to make it better readable. We deleted table 2A leaving table 2B as table 2. The reason is that 2A and 2B show the same information only in different ways.
  2. "The vast majority of children received neoadjuvant chemotherapy so the authors can not address the question as to whether there were fewer surgical complications with primary vs secondary surgery." This is correct. In this article we did not analyse complications of surgery.
  3. "The "bottom line" conclusions of this review are that patients with vena caval tumor thrombus had a higher rate of survival if they did not have diffuse anaplasia histology or metastatic disease.  These findings are certainly not at all surprising.  This study supports the finding in prior studies, but does not really provide new information regarding this clinical situation. " This statement is correct. But our analysis defined the two main factors in patients with VCT who do need more intensive treatment. In a multivariate COX regression VCT is no longer a rsik factor for survival, if diffuse anaplasia and no remission of metastasis after chemotherapy and surgery is achieved. But the group of patients with non-anaplastic tumors and complete remission of metastasis after preoperative chemotherapy and surgery are doing significantly better than those with diffuse anaplasia and missing complete remission in case of VCT (see also figure 1D).

Reviewer 2 Report

This retrospective study identified over 3000 children with Wilms tumor treated on historical GPOH studies, and identified 148 (5%) who developed a tumor thrombus extending to the inferior vena cava.  The investigators define the prevalence of tumor response to neoadjuvant chemotherapy, characterize features more likely associated with failure to respond to neoadjuvant chemotherapy, and suggest that this clinical finding is in itself a high risk finding.  This is a large and valuable data set, and although shares findings from prior studies of comparable size, is quite contemporary thereby provides an important update.  I do have several comments that I believe are necessary to address before the manuscript is suitable for consideration of publication.

Main Comments:

1. As highlighted in the final sentences concluding this manuscript, a general theme of the paper is that patients with VCT have a worse survival than those without VCT.  As shown in Table S1, most patients in the comparison (non-thrombosis) group are pathological stage 1 whereas very few in the thrombosis group are low stage whereas nearly half of the thrombosis study group have metastatic disease compared with much lower percentage in the comparison.  Do the authors believe that their findings regarding the worse prognosis in VCT are independent of stage (also tumor size, age, etc.)?  That is, does this finding reflect a worse biology informed by the VCT (perhaps a more invasive phenotype) or does this reflect the already known worse survival of patients with higher stage disease (i.e. that is, VCT is not an independent risk factor)?  A multivariate regression or even a selected comparison of just stage III and IV patients might be helpful to better address this.

2. For the commentary in lines 134-153 and 264-265 regarding the degree of shortening the thrombus with preoperative chemotherapy, it would be valuable to stratify this by the presence or absence of doxorubicin in the neoadjuvant chemotherap.  In particular, in non-anaplastic cases was preoperative chemotherapy with doxorubicin (as would be given for metastatic cases) more effective in shrinking the tumor thrombus.  This may help inform whether certain investigators might consider augmenting to AVD from AV in patients with very large thromboses.  Related, do the authors believe that there are some patients with IVC thromboses beyond a certain point (e.g. suprahepatic, intracardiac) who warrant addition of doxorubicin?

3. Why do the authors believe that preoperative chemotherapy resulted in only 23% shrinkage in their study as opposed to higher rates of shrinkage in previously published studies?

4. Since this may be a relevant consideration, authors should comment on whether anticoagulation was used, was not used, or is not known.

5. In the last paragraph of the discussion, the authors state, “Taking histology and metastatic disease into consideration, especially those patients with diffuse anaplasia and without remission of metastasis after PC are a high-risk group for treatment failure. This is a new finding.”  I’m not clear on this conclusion.  Diffuse anaplasia, metastatic disease [with any histology], and/or incomplete resolution of pulmonary metastatic disease (e.g. AREN0533) are not newly defined risk factors.  Are the authors concluding that this is a new finding, specifically in the context of IVC thromboses?  Seems like they are stating that this is new information more generally but maybe that is my misinterpretation.  Please clarify.

Minor Comments:

6. For the one patient with progressive thrombus, was it diffusely anaplastic or blastemal predominant histology?  If not, anything unusually high risk about this case?

7. Was there a concordance between completely necrotic histology (or regressive histology) and radiographic resolution of IVC thrombosis?

8. Line 136 states “To assess the effect of chemotherapy on tumor thrombus, 113 pairs of data with radiologic imaging could be used.”  This seems to suggest it wasn’t actually done when the authors did do this investigation.

9. Line 155 states, “Worsening 155 (larger, ) means increase of thrombus size by 1 level in one case.”  I believe that there is a punctuation error here.

10. Very minor point but just to confirm, although uncommon, we have seen patients with WT with contralateral thromboses (that is, tumor/LN compression was so large that it caused stasis in the opposite kidney with resultant bland thrombosis).  Can the authors confirm that all of the thromboses in this study were tumor thromboses from an affected kidney rather than a contralateral thrombus?

11. Please reconcile the discordance in the denominator of patients with VCT resolution in Table 2B (14 of 118) and line 143-144 (14 of 113)

12. Paragraph formatting at the end of line 223 should be corrected.

13. Suggest rewording line 200 from ‘vena cava inferior’ to ‘inferior vena cava’

14. Line 237-239, Suggest rewording “In almost every second patient (n=19/40) of their study, additional findings were found that had an impact on staging and therapy.”

Author Response

We thank the reviewer for very constructive comments that did improve our paper.

In detail we answer as follow:

ad 1.: The distribution of local stages between patients with and without VCT is significant different (stage I 12,2% vs 60,7%; stage II 36,7% vs 21.7%; stage III 51.0% vs 17.6%; p<0.001) as well as the rate of metastatic disease. In patients with metastatic disease 55,9% of patients (81/145) have metastases compared to 15,4% (446/2823.) Nevertheless, outcome between patients with or without VCT is the same if stages are taken into consideration. We have added a new paragraph and a new figure 1 showing these results. Because of higher local stages and more metastatic disease patients with VCT have a more aggresive tumor but do response to treatment in a way that outcome is not different. The worst outcome have those patients with diffuse anaplasia and no remission of metastasis after preoperative chemotherapy. Their overall survival (35 patients) is 77.1% after 5 years compared to 87.8% in those patients with no VCT (196 patients) (p=0.078) (new Figure 3).

ad 2.: This is a very good question. Altogether in 113 patients we could analyse the response of VCT. As we do not have data on the number of Doxorubicin given in each patient, we can only compare patients with and without metastatic disease on their response to doxorubicine differnetly. Of the 113 patients 64 had metastatic disease and 49 not. In metatatic patients 18 did respond (28.1%) and in non-metatstatic patients 11 did respond (22.4%). This is not statistically significant. We have added one sentence before table 2. Most importantly the histological type showed such a significant difference in response as already written.

ad 3.: The reason for this difference in response compared to literature is based on the way how we judged shrinkage. We did only measure the shrinkage as a change in the classification of the extend of the VCT and not in lenghth as in other papers, but not in all other studies it is explained how often VAD compared to VA was applied.

ad 4.: Thanks for this comment. Unfortunately, we do not have data about anticoagulation in our patient cohort.

ad 5.: We did see a difference in outcome between patients with and without VCT in this small patient group addressed. See also our answer to your comment 1.

ad 6.: This patient had diffuse anaplasia.

ad 7.: There was no correlation between completely necrotic histology (or regressive histology) and radiographic resolution of IVC thrombosis. One reason maybe that a VCT is composed of thrombus plus tumor and the thrombus may remain, even if the tumor is completely necrotic

ad 8.: These were all data from muticenter trials where missing data could not be avoided and because of the retrospective nature we could not get access to missing data, despite several attempts. Therefore response could only be analysed in those patients with available imaging dat at diagnosis and after preoperative chemotherapy.

ad 9.: Thanks for the comment. In response to reviewer 1, we have deleted table 2A as it did not contain more information than table 2B.

Ad 10.: We are aware of 1 patient having a thrombosis also in the renal vein of the healthy kidney and in addition a thrombus going down to the iliacal veins with blood congestion in the lower extremities.

Ad 11.: Altogether we have 118 patients with VCT and imaging studies. But in only 113 of these patients we have imaging studies at diagnosis and after preoperative chemotherapy. In the text only those 113 patients are considered where we could analyse the response to treatment. In table 2B (now table 2) all 118 patients are included. Nevertheless, as a response to reviewer 1 we did delete part of the text to avoid confusion by the reader.

Ad 12.: Thanks. ThIs is corrected.

Ad 13.: Thanks. We have changed accordingly.

Ad 14.: Thanks we have reworded this sentence to: "In almost every second patient (n=19/40) of their study, additional findings led to a change of local stage and treatment."